# Overlapping Community Detection with Graph Neural Networks

## Abstract

Community detection in graphs is of central importance in graph mining, machine learning and network science. Detecting *overlapping* communities is especially challenging, and remains an open problem. Motivated by the success of graph-based deep learning in other graph-related tasks, we study the applicability of this framework for overlapping community detection. We propose a probabilistic model for overlapping community detection based on the graph neural network architecture. Despite its simplicity, our model outperforms the existing approaches in the community recovery task by a large margin. Moreover, due to the inductive formulation, the proposed model is able to perform out-of-sample community detection for nodes that were not present at training time.

## 1 Introduction

Graphs provide a natural way of representing complex real-world systems. For understanding the structure and behavior of these systems, community detection methods are an essential tool. Detecting communities allows us to analyze social networks (Girvan & Newman, 2002), to detect fraud (Pinheiro, 2012), to discover functional units of the brain (Garcia et al., 2018), and to predict functions of proteins (Song & Singh, 2009). Over the last decades, this problem has attracted significant attention of the research community and numerous models and algorithms have been proposed (Xie et al., 2013). In particular, it is a well known fact that communities in real graphs are in fact overlapping (Yang & Leskovec, 2014), thus, requiring the development of advanced models to capture this complex structure.

In this regard, the advent of deep learning methods for graph-structured data opens new possibilities for designing more accurate and more scalable algorithms. Indeed, deep learning on graphs has already shown state-of-the-art results in s for various graph-related tasks such as semi-supervised node classification and link prediction (Cai et al., 2018). Likewise, a few deep learning methods for community detection in graphs have been proposed (Yang et al., 2016; Chen et al., 2017). However, they all have one drawback in common: they only focus on the special case of disjoint (non-overlapping) communities. Handling overlapping communities, is a requirement not yet met by existing deep learning approaches to community detection.

In this paper we propose an end-to-end deep probabilistic model for overlapping community detection in graphs. Our core idea lies in predicting the community affiliations using a graph neural network. Despite its simplicity, our model achieves state-of-the art results in community recovery and significantly outperforms the existing approaches. Moreover, our model is able to perform out-of-sample (inductive) community detection for nodes that were not seen at training time. To summarize, our main contributions are:

- We propose the Deep Overlapping Community detection (DOC) model - a simple, yet effective deep learning model for overlapping community detection in graphs. DOC is one of few methods is able to perform community detection both transductively and inductively.

- We introduce 5 new datasets for overlapping community detection, which can act as a benchmark to stimulate future work in this research area.

- We perform a thorough experimental evaluation of our model, and show its superior performance when comparing with established methods for overlapping community detection.

## 2 COMMUNITY DETECTION IN GRAPHS

Assume that we are given an undirected unweighted graph $\mathcal{G} = (\mathcal{V}, \mathcal{E})$, with $N := |\mathcal{V}|$ nodes and $M := |\mathcal{E}|$ edges, represented by a symmetric adjacency matrix $\boldsymbol{A} \in \{0, 1\}^{N \times N}$. Moreover, each node is associated with $D$ real-valued attributes, that can be represented as a matrix $\boldsymbol{X} \in \mathbb{R}^{N \times D}$. The goal of overlapping community detection is to assign nodes in the graph into $C$ communities. Such assignment can be represented as a non-negative community affiliation matrix $\boldsymbol{F} \in \mathbb{R}_{\geq 0}^{N \times C}$, where $F_{uc}$ denotes the strength of node $u$'s membership in community $c$ (with the notable special case of binary hard-assignment $\boldsymbol{F} \in \{0, 1\}^{N \times C}$). There is no single universally definition of community in the literature. However, most recent works tend to agree with the statement that a community is a group of nodes that have a higher probability to form edges with each other than with other nodes in the graph (Fortunato & Hric, 2016).

One can broadly subdivide the existing methods for overlapping community detection into three categories: approaches based on non-negative matrix factorization (NMF), probabilistic inference, or heuristics. Methods based on NMF are trying to recover the community affiliation matrix $\boldsymbol{F}$ by performing a low-rank decomposition of the adjacency matrix $\boldsymbol{A}$ or some other related matrix (Wang et al., 2011; Kuang et al., 2012). Probabilistic approaches, such as Yang & Leskovec (2013) or Zhou (2015), treat $\boldsymbol{F}$ as a latent variable in a generative model for the graph, $p(\boldsymbol{A}, \boldsymbol{F})$. This way the problem of community detection is cast as an instance of probabilistic inference. Lastly, heuristic-based approaches usually define a goodness measure, like within-community edge density (Galbrun et al., 2014), and then directly optimize it. All of these approaches can be very generally formulated as an optimization problem

$$\min_{\boldsymbol{F}} \mathcal{L}(\boldsymbol{F}) \qquad (1)$$

for an appropriate choice of the loss function $\mathcal{L}$, be it Frobenius norm $\mathcal{L}(\boldsymbol{F}) = \|\boldsymbol{A} - \boldsymbol{F}\boldsymbol{F}^T\|_F$ or negative log-likelihood $\mathcal{L}(\boldsymbol{F}) = -\log p(\boldsymbol{A}|\boldsymbol{F})$ of some generative model.

Besides these traditional approaches, one can also view the problem of community detection through the lens of representation learning. The community affiliation matrix $\boldsymbol{F}$ can be considered as an embedding of nodes into $\mathbb{R}_{\geq 0}^C$, with the aim of preserving the community structure. Given the recent success of deep representation learning for graphs (Cai et al., 2018), a question arises: "Can the advances in deep representation learning for graphs be used to design better community detection algorithms?".

A very simple idea is to first apply a node embedding approach to the graph, and then cluster the nodes in the embedding space using $k$-means to obtain communities (as done in, e.g., Tsitsulin et al. (2018)). However, such approach is only able to detect disjoint communities, which does not correspond to the structure of communities in real-world graphs (Yang & Leskovec, 2014). Instead, we argue that an end-to-end deep learning architecture able to detect overlapping communities is preferable.

Traditional community detection methods treat $\boldsymbol{F}$ as a free variable, with respect to which optimization is performed (Equation 1). This is similar to how embeddings are learned in methods like DeepWalk (Perozzi et al., 2014) and node2vec (Grover & Leskovec, 2016). In contrast, recent works of Kipf & Welling (2016); Hamilton et al. (2017); Bojchevski & Günnemann (2018) have adopted the approach of defining the embeddings as a function of node attributes $\boldsymbol{F} := f_{\boldsymbol{\theta}}(\boldsymbol{X}, \boldsymbol{A})$ and solving the optimization problem

$$\min_{\boldsymbol{\theta}} \mathcal{L}(f_{\boldsymbol{\theta}}(\boldsymbol{X}, \boldsymbol{A})), \qquad (2)$$

where $f_{\boldsymbol{\theta}}$ is defined by a neural network. [1] Such formulation allows to

- achieve better performance in downstream tasks like link prediction and node classification;
- naturally incorporate the attributes $\boldsymbol{X}$ without hand-crafting the generative model $p(\boldsymbol{X}, \boldsymbol{F})$;
- generate embeddings inductively for previously unseen nodes.

We propose to use this framework for overlapping community detection, and describe our model in the next section.

---

[1] Even if node attributes are not available, this approach can work by, for example, setting $\boldsymbol{X}$ to an identity matrix $\boldsymbol{I}$ or to the adjacency matrix $\boldsymbol{A}$.

## 3 THE DOC MODEL

We let the community affiliations be produced by a three-layer graph convolutional neural network (GCN), as defined in Kipf & Welling (2017).

$$\boldsymbol{F} = \mathrm{GCN}_{\boldsymbol{\theta}}(\boldsymbol{X}, \boldsymbol{A}) = \mathrm{ReLU}(\hat{\boldsymbol{A}}\,\mathrm{ReLU}(\hat{\boldsymbol{A}}\,\mathrm{ReLU}(\hat{\boldsymbol{A}}\boldsymbol{X}\boldsymbol{W}^{(0)})\boldsymbol{W}^{(1)})\boldsymbol{W}^{(2)}), \tag{3}$$

where $\hat{\boldsymbol{A}} = \tilde{\boldsymbol{D}}^{-\frac{1}{2}}\tilde{\boldsymbol{A}}\tilde{\boldsymbol{D}}^{-\frac{1}{2}}$ is the normalized adjacency matrix, $\tilde{\boldsymbol{A}} = \boldsymbol{A}+\mathbf{I}$, and $\tilde{\boldsymbol{D}}$ the corresponding degree matrix. A ReLU nonlinearity is applied element-wise to the output layer to ensure non-negativity of the community affiliation matrix $\boldsymbol{F}$. Any other graph neural network architecture can be used here – we choose GCN because of its simplicity and popularity.

**Link function.** A good $\boldsymbol{F}$ explains well the community structure of the graph. To model this formally, we adopt a probabilistic approach to community detection where we need to define the likelihood $p(\boldsymbol{A}|\boldsymbol{F})$. A standard assumption in probabilistic community detection is that the edges $A_{uv}$ are conditionally independent given the community memberships $\boldsymbol{F}$. Thus, once the $\boldsymbol{F}$ matrix is given, every pair of nodes $(u, v)$ produces an interaction based on their community affiliations $X_{uv} = \boldsymbol{F}_u\boldsymbol{F}_v^T \in \mathbb{R}_{\geq 0}$. For a probabilistic interpretation, this interaction is transformed into an edge probability by means of a link function $g : \mathbb{R}_{\geq 0} \to [0, 1]$. The edge probability is then given by

$$A_{uv} \sim \mathrm{Bern}(g(\boldsymbol{F}_u\boldsymbol{F}_v^T)) \tag{4}$$

We consider two choices for the link function $g$: Bernoulli-Poisson link and sigmoid link.

**Bernoulli-Poisson link**, defined as $\xi(X_{uv}) = 1 - \exp(-X_{uv})$, is a common probabilistic model for overlapping community detection (Yang & Leskovec, 2013; Zhou, 2015; Todeschini et al., 2016). Note, that under the BP model a pair of nodes that have no communities in common (i.e. $\boldsymbol{F}_u\boldsymbol{F}_v^T = 0$) have a zero probability of forming an edge. This is an unrealistic assumption, which can be easily fixed by adding a small offset $\varepsilon > 0$, that is $\xi(X_{uv}) = 1 - \exp(-X_{uv} - \varepsilon)$.

**Sigmoid link**, defined as $\sigma(x) = (1 + \exp(-X_{uv}))^{-1}$, is the standard choice for binary classification problems. It can also be used to convert the edge scores into probabilities in probabilistic models for graphs (Latouche et al., 2011; Tang et al., 2015; Kipf & Welling, 2016). Since a non-negative $\boldsymbol{F}$ implies that the interactions between every pair of nodes $X_{uv}$ are at least 0, the edge probability under the sigmoid model is always above $\sigma(0) = 0.5$. This can be fixed by introducing an offset: $g(X_{uv}) = \sigma(X_{uv} - b)$. The offset $b$ becomes an additional variable to optimize over, closed-form expression for which is provided in Schein et al. (2003). However, while optimizing over $b$ produces better likelihood scores, we have empirically observed that fixing it to zero leads to the same performance in community recovery (Section 4.3). Thus, we set $b = 0$ in our experiments.

We consider both link functions, and denote the two variants of our model as DOC-BP and DOC-Sigmoid for Bernoulli-Poisson and sigmoid link functions respectively.

**Loss.** Once the link function $g$ is chosen, the Equation 4 defines the likelihood function $p(\boldsymbol{A}|\boldsymbol{F})$. Maximizing the likelihood is equivalent to minimizing the negative log-likelihood, which corresponds to the well-known binary cross-entropy loss function. Since real-world graphs are extremely sparse (only $10^{-2} - 10^{-5}$ of possible edges are present), we are dealing with an extremely imbalanced binary classification problem. A standard way of dealing with this problem is by balancing the contribution from both classes, which corresponds to the following objective function

$$\mathcal{L}(\boldsymbol{F}) = -\mathbb{E}_{(u,v)\sim P_E}\left[\log p(A_{uv} = 1|\boldsymbol{F}_u, \boldsymbol{F}_v)\right] - \mathbb{E}_{(u,v)\sim P_N}\left[\log p(A_{uv} = 0|\boldsymbol{F}_u, \boldsymbol{F}_v)\right], \tag{5}$$

where $P_E$ and $P_N$ stand for uniform distributions over edges and non-edges respectively.

Evaluating the gradient of the full loss requires $\mathcal{O}(N^2)$ operations (since we need to compute the expectation over $\binom{N}{2}$ possible edges/non-edges). This is impractical even for moderately-sized graphs. Instead, we optimize the objective using stochastic gradient descent. That is, at every iteration we approximate $\nabla\mathcal{L}$ using $S$ randomly sampled edges, as well as the same number of non-edges.

To summarize, we use stochastic gradient descent to optimize the objective

$$\min_{\boldsymbol{\theta}} \mathcal{L}(\mathrm{GCN}_{\boldsymbol{\theta}}(\boldsymbol{X}, \boldsymbol{A})), \tag{6}$$

where the parameters $\boldsymbol{\theta}$ are the weights of the neural network, $\{\boldsymbol{W}^{(0)}, \boldsymbol{W}^{(1)}, \boldsymbol{W}^{(2)}\}$.

Table 1: Dataset statistics. $K$ stands for 1000.

| Dataset | Network type | $N$ | $M$ | $D$ | $C$ |
|---------|-------------|-----|-----|-----|-----|
| Facebook-0 | Social | $0.3K$ | $2.5K$ | 30 | 24 |
| Facebook-107 | Social | $1.0K$ | $26.7K$ | 11 | 9 |
| Facebook-1684 | Social | $0.8K$ | $14.0K$ | 15 | 17 |
| Facebook-1912 | Social | $0.8K$ | $30.0K$ | 29 | 46 |
| Facebook-3437 | Social | $0.5K$ | $4.8K$ | 23 | 32 |
| Amazon | Co-purchase | $7.7K$ | $245.2K$ | $4.1K$ | 14 |
| Coauthor-CS | Co-authorship | $18.3K$ | $81.9K$ | $6.8K$ | 15 |
| Coauthor-Physics | Co-authorship | $34.5K$ | $248.0K$ | $8.4K$ | 5 |
| Reddit-Gaming | Online discussion | $23.1K$ | $346.8K$ | $7.6K$ | 48 |
| Reddit-Technology | Online discussion | $10.8K$ | $162.7K$ | $7.1K$ | 31 |

## 4 EVALUATION

### 4.1 PRELIMINARIES

**Datasets.** We perform all our experiments using the following real-world graph datasets. **Facebook** (Mcauley & Leskovec, 2014) is a collection of small (100-1000 nodes) ego-networks from the Facebook graph. In our experiments we consider the 5 largest of these ego-networks (Facebook-0, Facebook-107, Facebook-1684, Facebook-1912, Facebook-3437).

Larger graph datasets (1000+ nodes) with reliable ground-truth overlapping community information, and node attributes are not openly available, which hampers the evaluation of methods for overlapping community detection in attributed graphs. For this reason we have collected and pre-processed 5 real-world datasets, that satisfy these criteria and can act as future benchmarks (we will provide the datasets for download after the blind-reviewing phase). **Coauthor-CS** and **Coauthor-Physics** are subsets of the Microsoft Academic co-authorship graph, constructed based on the data from the KDD Cup 2016[2]. Communities correspond to research areas in computer science and physics respectively. **Reddit-Technology** and **Reddit-Gaming** represent user-user graphs from the content-sharing platform Reddit[3]. Communities correspond to subreddits – topic-specific communities that users participate in. **Amazon** is a segment of the Amazon co-purchase graph (McAuley et al., 2015), where product categories represent the communities. Details about how the datasets were constructed and exploratory analysis are provided in Appendix B.

**Model architecture.** We denote the model variant with the Bernoulli-Poisson link as DOC-BP, and the model variant with the sigmoid link as DOC-Sigmoid. For all experiments we use a 3-layer GCN (Equation 3) as the basis for both models. We use the same model configuration for all other experiments, unless otherwise specified. More details about the model and the training procedure are provided in Appendix A. All reported results are averaged over 10 random initializations, unless otherwise specified.

### 4.2 CONVERGENCE OF THE STOCHASTIC SAMPLING

As mentioned in Section 3, evaluation of the full loss (Equation 5) and its gradients is computationally prohibitive due to its $\mathcal{O}(N^2)$ scaling. Instead, we propose to use a stochastic approximation, that only depends on the fixed batch size $S$. We perform the following experiment to ensure that our training procedure converges to the same result, as when using the full objective.

**Experimental setup.** We train the the two variants of the model on the Facebook-1912 dataset, since it is small enough ($N = 755$) for full-batch training to be feasible. We compare the full-batch training procedure with stochastic training for different choices of the batch size $S$. Starting with the same initialization, we measure the respective full losses (Equation 5) over the iterations.

---

[2] https://kddcup2016.azurewebsites.net/
[3] https://files.pushshift.io/reddit/

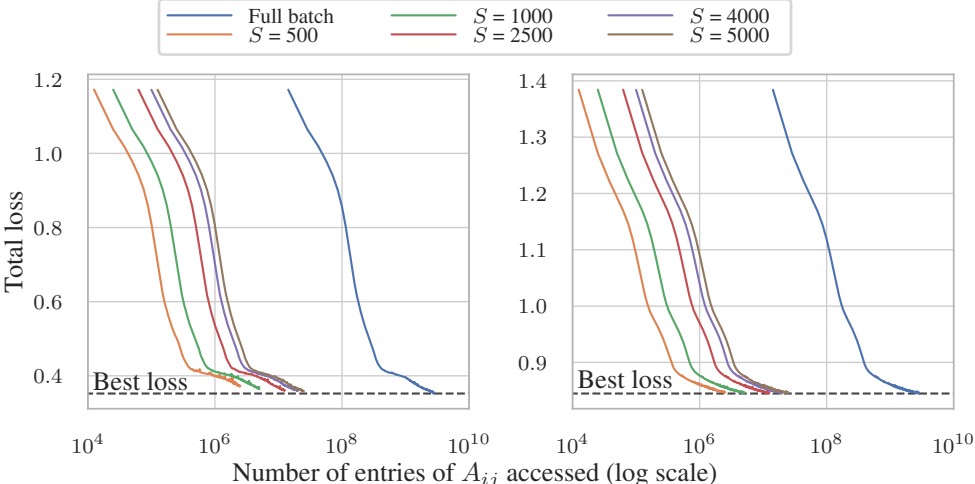

Figure 1: Training curves for DOC-BP (left) and DOC-Sigmoid (right) models. For both models and for all batch sizes, the model converges to a solution of the same quality, as when using full-batch training.

**Results.** Figure 1 shows training curves for batch sizes $S \in \{500, 1000, 2500, 4000, 5000\}$, as well as for full-batch training. As we see, the stochastic training procedure is stable. For all batch sizes the loss converges very closely to the value achieved by full-batch training.

### 4.3 RECOVERY OF GROUND-TRUTH COMMUNITIES

The standard way of comparing overlapping community detection algorithms is by assessing how well they can recover communities in graphs, for which the ground truth community affiliations are known. It may happen that the information used as "ground truth communities" does not correlate with the graph structure (Peel et al., 2017). For the datasets considered in this paper, however, ground truth communities make sense both intuitively and quantitatively (see Appendix B for a more detailed discussion). Therefore, good performance in this experiment is a good indicator of the utility of an algorithm.

**Predicting community membership.** In order to compare the detected communities to the ground truth, we first need to convert continuous community affiliations $\boldsymbol{F}$ into binary community assignments. We assign node $u$ to community $c$ if $F_{uc}$ is above a threshold $\rho$. We set $\rho = 0.4$ for DOC-BP and $\rho = 0.2$ for DOC-Sigmoid, as these are the values that achieve the best performance on the Facebook-1912 dataset.

**Metrics.** We use overlapping normalized mutual information (NMI), as defined by McDaid et al. (2011), in order to quantify the agreement of the detected communities with the ground-truth data.

**Baselines.** We compare our method against a number of established methods for overlapping community detection. BigCLAM (Yang & Leskovec, 2013) is a probabilistic model based on the Bernoulli-Poisson link that only considers the graph structure. CESNA (Yang et al., 2013) is an extension of BigCLAM, that additionally models the generative process for node attributes. SNMF (Kuang et al., 2012) and CDE (Li et al., 2018) are non-negative matrix factorization approaches for overlapping community detection. We also compared against the LDense algorithm from Galbrun et al. (2014) – a heuristic-based approach, that finds communities with maximum edge density and similar attributes. However, since it achieved less than $1\%$ NMI for 8 out of 10 datasets, we don't include the results for LDense into the table. To ensure a fair comparison, all methods were given the true number of communities $C$. Other hyperparameters were set to their recommended values. Detailed configurations of the baselines are provided in Appendix C.

**Results.** Table 2 shows how well different methods score in recovery of ground-truth communities. DOC-BP achieves the best or the second best score for 9 out of 10 datasets. DOC-Sigmoid achieves

Table 2: Recovery of ground-truth communities, as measured by normalized mutual information (in %). All results are averaged over 10 initializations. Best result for each row in **bold**, second best underlined. N/A - method not applicable to this attribute type. D.N.F. - did not finish in 24 hours.

|  | BigCLAM | CESNA | SNMF | CDE | DOC-BP | DOC-Sigmoid |
|---|---|---|---|---|---|---|
| Facebook-0 | 4.6 | 5.4 | 3.1 | 6.6 | 10.4 | **12.3** |
| Facebook-107 | **9.8** | 9.2 | 6.1 | 6.3 | 8.2 | 9.2 |
| Facebook-1684 | 32.7 | 28.0 | 13.0 | 28.8 | 34.8 | **36.3** |
| Facebook-1912 | 21.4 | 21.2 | 23.3 | 15.5 | 37.7 | **41.2** |
| Facebook-3437 | 0.0 | 0.0 | 0.0 | 0.0 | **1.8** | 1.5 |
| Amazon | 22.2 | N/A | 13.3 | D.N.F. | 29.1 | **30.6** |
| Coauthor-CS | 2.6 | 34.5 | 10.2 | D.N.F. | **44.1** | 38.0 |
| Coauthor-Physics | 3.7 | 17.5 | 8.2 | D.N.F. | **38.8** | 30.6 |
| Reddit-Gaming | 35.8 | 37.8 | 17.8 | D.N.F. | 40.6 | **42.3** |
| Reddit-Technology | 14.4 | 23.8 | 19.2 | D.N.F. | **37.7** | 30.0 |

the best or the second best score 10 out of 10 times. This demonstrates the potential of deep learning methods for overlapping community detection. CESNA could not be run for the Amazon dataset, because it cannot handle continuous attributes. In contrast, both DOC model variants can be used with any kind of attributes out of the box. CDE was not able to process any of the graphs with $N \geq 7K$ nodes within 24 hours. On the other hand, both DOC-BP and DOC-Sigmoid converged in 30s-6min for all datasets except Amazon, where it took up to 20 minutes because of the dense attribute matrix.

### 4.4 Do we really need a graph neural network?

As we just saw, the DOC-BP and DOC-Sigmoid models, both based on the GCN architecture, are able to achieve superior performance in community detection. Intuitively, it makes sense to use a graph neural network (GNN) in our setting, since it allows to incorporate the attribute information and also produces similar community vectors, $F_u$, for adjacent nodes. Nevertheless, we should ask whether it's possible achieve comparable results with a simpler model. To answer this question, we consider the following two baselines.

**Multilayer perceptron (MLP):** Instead of a GCN (Equation 3), we use a simple fully-connected neural network to generate $F$.

$$F = \text{MLP}_{\boldsymbol{\theta}}(\boldsymbol{X}) = \text{ReLU}(\text{ReLU}(\text{ReLU}(\boldsymbol{X}\boldsymbol{W}^{(0)})\boldsymbol{W}^{(1)})\boldsymbol{W}^{(2)}) \tag{7}$$

This is indeed related to the model proposed by Hu et al. (2017). For this baseline, we use the same configuration (number and sizes of layers, training procedure, etc.) as for the GCN-based model. Same as for GCN (Equation 6), we optimize the parameters of the MLP, $\boldsymbol{\theta} = \{\boldsymbol{W}^{(0)}, \boldsymbol{W}^{(1)}, \boldsymbol{W}^{(2)}\}$, using stochastic gradient descent.

$$\min_{\boldsymbol{\theta}} \mathcal{L}(\text{MLP}_{\theta}(\boldsymbol{X})) \tag{8}$$

**Free variable (FV):** As an even more simple baseline, we consider treating the community affiliations $F$ as a free variable in optimization.

$$\min_{\boldsymbol{F} \geq 0} \mathcal{L}(\boldsymbol{F}) \tag{9}$$

This is similar to standard community detection methods like BigCLAM. Since this optimization problem is rather different from those of GCN (Equation 6) and MLP (Equation 8), we perform additional hyperparameter optimization for the FV model. We consider different choices for the learning rate and two initialization strategies, while keeping other aspects of the training procedure as before (stochastic training, early stopping). We pick the configuration that achieved the best average NMI score across all datasets. Note that this gives a strong advantage to the FV model, since for GCN and MLP models the hyperparameters were fixed without the knowledge of the ground-truth communities.

**Experimental setup.** We compare the NMI scores obtained by all three models, both for Bernoulli-Poisson and sigmoid link functions.

Table 3: Comparison of graph neural network (GNN) based model against simpler baselines in the ground-truth community recovery task. Multilayer perceptron (MLP) and Free Variable (FV) models are optimizing the same objective (Equation 5), but represent the community affiliations $\boldsymbol{F}$ differently.

|  | Bernoulli-Poisson link | | | Sigmoid link | | |
|---|---|---|---|---|---|---|
|  | FV | MLP | GNN | FV | MLP | GNN |
| Facebook-0 | 5.1 | 1.6 | **10.4** | 5.5 | 0.5 | **12.3** |
| Facebook-107 | **13.2** | 4.3 | 8.2 | 9.0 | 8.3 | **9.2** |
| Facebook-1684 | **39.0** | 4.6 | 34.8 | 26.0 | 8.0 | **36.3** |
| Facebook-1912 | 21.3 | 3.4 | **37.7** | 27.0 | 7.8 | **41.2** |
| Facebook-3437 | 0.0 | 1.2 | **1.8** | 0.3 | 0.4 | **1.5** |
| Amazon | 9.1 | 0.1 | **29.1** | 20.3 | 0.7 | **30.6** |
| Coauthor-CS | 0.0 | 26.3 | **44.1** | 12.7 | **38.7** | 38.0 |
| Coauthor-Physics | 0.0 | 35.0 | **38.8** | 9.0 | **31.1** | 30.6 |
| Reddit-Gaming | 33.1 | 0.0 | **40.6** | 28.7 | 6.0 | **42.3** |
| Reddit-Technology | 20.9 | 5.0 | **37.7** | 27.6 | 11.2 | **30.0** |

**Results.** As shown in Table 3, GNN-based models outperforms the simpler baselines in 16 out of 20 cases (Remember, that the free variable version even had the advantage of picking the hyperparmeters that lead to the highest NMI scores). This highlights the fact that attribute information only is not enough for community detection, and incorporating the graph structure clearly helps to make better inferences.

## 4.5 Out-of-sample community detection

So far, we have observed that the DOC model is able to recover communities with high precision. What's even more interesting, since our model learns the mapping from node attributes to the producing community affiliations (Equation 3), it should also be possible to predict communities inductively for nodes that were not present at training time.

**Experimental setup.** We hide a randomly selected fraction of nodes from each community, and train the DOC-BP and DOC-Sigmoid models on the remaining graph. Once the parameters $\boldsymbol{\theta}$ are learned, we perform a forward pass of each model using the full adjacency and attribute matrix. We then compute how well the communities were predicted for the nodes that were not present during training, using NMI as a metric. We compare with the MLP model (Equation 7) as a baseline.

**Results.** As can be seen in Figure 2, both DOC-BP and DOC-Sigmoid are able to infer communities inductively for previously unseen nodes with high accuracy (NMI $\geq 40\%$), which is on the same level as for the transductive setting (Table 3). On the other hand, MLP-BP and MLP-Sigmoid models both perform worse than the GCN-based ones, and significantly below their own scores for transductive community detection. This highlights the fact that graph-based neural network architectures provide a significant advantage for community detection.

## 5 Related work

The problem of community detection in graphs is well-established in the research community, and methods such as stochastic block models (Abbe, 2018) and spectral methods (Von Luxburg, 2007) have attracted a lot of attention. Despite the popularity of these methods, they are only suitable for detecting non-overlapping communities (i.e. partitioning the network), which is not the setting usually encountered in real-world networks (Yang & Leskovec, 2014). Methods for overlapping community detection have been proposed (Xie et al., 2013), but our understanding of their behavior is not as mature as for the non-overlapping methods.

As discussed in Section 2, methods for OCD can be broadly divided into methods based on nonnegative matrix factorization, probabilistic inference and heuristics. These categories are not mutually exclusive, and often one method can be viewed as belonging to multiple categories. For example, the factorization-based approaches that minimize the Frobenius norm $\|\boldsymbol{A} - \boldsymbol{F}\boldsymbol{F}^T\|_F$ (Wang

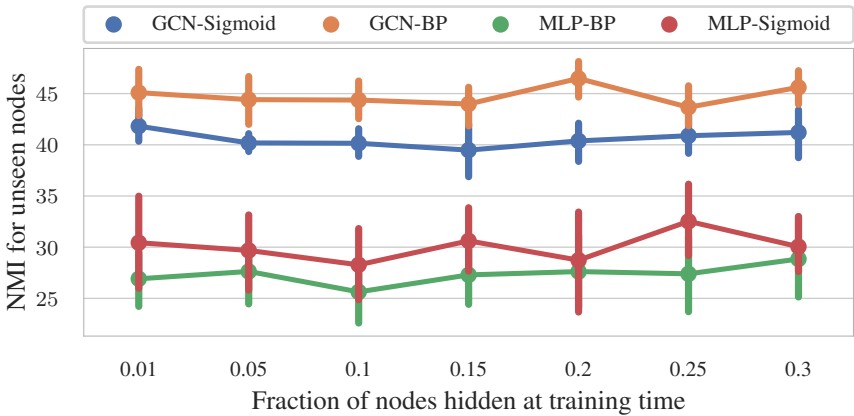

Figure 2: Out-of-sample community detection results of DOC-BP and DOC-Sigmoid.

et al., 2011) can be viewed as performing maximum likelihood estimation under a Gaussian model $p(\boldsymbol{A}|\boldsymbol{F}) = \prod_{u,v} \mathcal{N}(A_{uv}|\boldsymbol{F}_u \boldsymbol{F}_v^T, 1)$. More generally, most NMF and probabilistic inference models are performing a non-linear low rank decomposition of the adjacency matrix, which can be connected to the generalized principle component analysis model (Collins et al., 2002).

Deep learning for graph data can be broadly subdivided into two main categories: graph neural networks and node embeddings. Graph neural networks (Kipf & Welling, 2017; Hamilton et al., 2017) are specialized neural network architectures that can operate on graph-structured data. The goal of embedding approaches (Perozzi et al., 2014; Grover & Leskovec, 2016; Bojchevski & Günnemann, 2018) is to learn vector representations of nodes in a graph, that can later be used for other downstream machine learning tasks. One can perform $k$-means clustering on the node embeddings (as done in, e.g., Tsitsulin et al. (2018)) to cluster nodes into communities. However, such approach is not able to capture the overlapping community structure present in real-world graphs.

Several works have devised deep learning methods for community detection in graphs. Yang et al. (2016) and Cao et al. (2018) propose deep learning approaches that seek a low-rank decomposition of the modularity matrix (Newman, 2006). This means both of these approaches are limited to finding disjoint communities, as opposed to our algorithm. Also related to our model is the approach by Hu et al. (2017), where they use a deep belief network to generate the community affiliation matrix. However, their neural network architecture does not use the graph, which we have shown to be crucial in Section 4.4. Lastly, Chen et al. (2017) designed a neural network architecture for supervised community detection. Their model learns to detect communities by training on a labeled set with community information given. This is very different from this paper, where we learn to detect communities in a fully unsupervised manner.

## 6  DISCUSSION & CONCLUSIONS

In this work we have proposed and studied two deep models for overlapping community detection: DOC-BP, based on the Bernoulli-Poisson link, and DOC-Sigmoid, that relies on the sigmoid link function. The two variants of our model achieve state-of-the-art results and convincingly outperfom existing techniques in transductive and inductive community detection tasks. Using stochastic training, both approaches are highly efficient and scale to large graphs. Among the two proposed models, DOC-BP one average performs better than the DOC-Sigmoid variant. We leave to future work to investigate the properties of communities detected by these two methods.

To summarize, the results obtained in our experimental evaluation provide strong evidence that deep learning for graphs deserves more attention as a framework for overlapping community detection.

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

## A  MODEL CONFIGURATION

**Architecture.** We use a 3-layer graph convolutional neural network (Equation 3), with hidden sizes of 64, and the final (third) layer has size $C$ (number of communities to be detected). Dropout with 50% keep probability is applied at every layer. We don't use any other forms of regularization, such as weight decay.

**Training.** We train the model using Adam optimizer with default parameters. The learning rate is set to $10^{-4}$. We use the following early stopping strategy: Before training, we set aside 1% of edges and the same number of non-edges. Every 25 gradient steps we compute the loss (Equation 5) for the validation edges and non-edges. We stop training if there was no improvement to the best validation loss for $20 \times 25 = 500$ iterations, or after 5000 epochs, whichever happens first.

## B  DATASETS

### B.1  AMAZON

Raw Amazon data is provided by McAuley et al. (2015) at `http://jmcauley.ucsd.edu/data/amazon/links.html`.

- *Nodes:* A node in the graph represents a product sold at Amazon. To get a graph of manageable size, we restrict our attention to the products in 14 randomly-chosen subcategories of the "Video Games" supercategory.
- *Edges:* A pair of products $(u, v)$ is connected by an edge if $u$ is "also bought" with product $v$, or the other way around. The "also bought" information is provided in the raw data.
- *Communities:* We treat the subcategories (e.g. "Xbox 360", "PlayStation 4") as community labels. Every product can belong to multiple categories.
- *Features:* The authors of He & McAuley (2016) extracted visual features from product pictures using a deep CNN. We use these visual features as attributes $\boldsymbol{X}$ in our experiments.

### B.2  COAUTHOR (MICROSOFT ACADEMIC GRAPH)

We use the dump of the Microsoft Academic Graph that was published for the KDD CUP 2016 competition (`https://kddcup2016.azurewebsites.net/`) to construct two co-authorship graphs – Coauthor-CS and Coauthor-Physics.

- *Nodes:* A node in the graph represents a researcher.
- *Edges:* A pair of researchers $(u, v)$ is connected by an edge if $u$ and $v$ have co-authored one or more papers.
- *Communities:* For Computer Science (Coauthor-CS), we use venues as proxies for fields of study. We pick top-5 venues for each subfield of CS according to Google Scholar (`scholar.google.com`). An author $u$ is assigned to field of study $c$ if he published at least 5 papers in venues associated with this field of study.

  For Physics (Coauthor-Physics), we use the Physical Review A, B, C, D, E journals as indicators of fields of study (= communities). An author $u$ is assigned to field of study $c$ if he published at least 5 papers in the respective Physical Review "?" journal.
- *Features:* For author user $u$ we construct a histogram over keywords that were assigned to their papers. That is, the entry of the attribute matrix $X_{ud}$ = # of papers that author $u$ has published that have keyword $d$.

For this graph we had to remove from our consideration the papers that had too many ($\geq 40$) authors, since it led to very large fully-connected components in the resulting graph.

### B.3  REDDIT

Reddit is an online content-sharing platform, where users share, rate, and comment on content on a wide range of topics. The site consists of a number of smaller topic-oriented communi-

ties, called subreddits. We downloaded a dump of Reddit comments for February 2018 from `http://files.pushshift.io/reddit/comments/`. Using the list provided at `https://www.reddit.com/r/ListOfSubreddits/`, we picked 48 gaming-related subreddits and 31 technology-based subreddits. For each of these groups of subreddits we constructed a graph as following:

- *Nodes:* A node in the graph represents a user of Reddit, identified by their `author_id`.
- *Edges:* A pair of users $(u, v)$ is connected by an edge if $u$ and $v$ have both commented on the same 3 or more posts.
- *Communities:* We treat subreddits as communities. A user is assigned to a community $c$, if he commented on at least 5 posts posted in that community.
- *Features:* For every user $u$ we construct a histogram of other subreddits (excluding the subreddits used as communities) that they commented in. That is, the entry of the attribute matrix $X_{ud} = $ # of comments user $u$ left on subreddit $d$.

### B.4    EXPLORATORY ANALYSIS

Co-purchase graphs, co-authorship graphs and content-sharing platforms are classic examples of networks with overlapping community structure (Yang & Leskovec, 2014), so using these communities as ground truth is justifiable. Additionally, we show that for all the five graphs considered, the probability of connection between a pair of nodes grows monotonically with the number of shared communities. This further shows that our choice of communities makes sense.

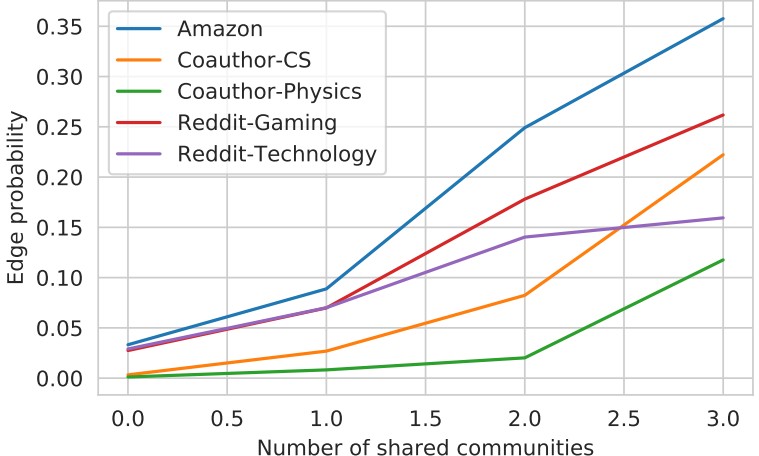

Figure 3: Edge probabilities for different sizes of shared communities for all of the constructed datasets.

## C    BASELINES

- We used the reference C++ implementations of BigCLAM and CESNA, that were provided by the authors (`https://github.com/snap-stanford/snap`). Models were used with the default parameter settings for step size, backtracking line search constants, and balancing terms. Since CESNA can only handle binary attributes, we binarize the original attributes (set the nonzero entries to 1) if they have a different type.
- We implemented SNMF ourselves using Python. The $\boldsymbol{F}$ matrix is initialized by sampling from the $\mathrm{Uniform}[0, 1]$ distribution. We run optimization until the improvement in the reconstruction loss goes below $10^{-4}$ per iteration, or for 300 epochs, whichever happens first.

- We use the Matlab implementation of CDE provided by the authors. We set the hyperparameters to $\alpha = 1$, $\beta = 2$, $\kappa = 5$, as recommended in the paper, and run optimization for 20 iterations.
- We use the Python implementation of LDense provided by the authors (`https://research.cs.aalto.fi/dmg/software.shtml`), and run the algorithm with the recommended parameter settings. Same as with CESNA, since the methods only supports binary attributes, we binarize the original data if necessary.

