# OpenReview forum: "Overlapping Community Detection with Graph Neural Networks"
_ICLR.cc/2019/Conference_

### Official Review · AnonReviewer1 · 2018-10-31
**Interesting combination of neural nets and network models, but not solid.**

**Rating:** 4
**Confidence:** 5

**Review:**

This paper proposes Deep Overlapping Community detection model (DOC), a graph convolutional network (GCN) based community detection algorithm for network data. The model is a simple combination of GCN and existing framework for community detection. The proposed algorithm is compared to baselines on various datasets, and demonstrated to be accurate in many cases.

I think the paper does not deal with one of the most important aspects of network modeling - the degree heterogeneity of nodes. Many works reported that lack of degree corrections would result in bad estimates of community structures [1,2,3]. Probably including the degrees as feature of nodes would be helpful.

Regarding the stochastic gradient descent by edge subsampling, I think the authors should mention [4], where the idea of edge subsampling in stochastic gradient descent setting was introduced before this work. Also, it is worth noting that we may lose some important distributional properties in graphs if we naively subsample from it [5]. For instance, sampling from positive and negative pairs to balance the class contribution may distort the sparsity and degree distributions of subsampled graphs.

If we choose to use Bernoulli-Poisson link function, we can reduce the time complexity of likelihood and gradient computation to O(N + E), where N is the number of nodes and E is the number of edges, with the auxiliary variable trick introduced in [6]. In that case we don't really have to worry about subsampling. Why didn't you consider applying this to your model?

Regarding the experiments, I think some important baselines are missing [3, 6]. Also, I wonder whether the proposed algorithm would scale to the graphs with more than 100,000 nodes.

References
[1] B. Karrer and M. E. J. Newman. Stochastic blockmodels and community structure in networks. Physical Review E, 83(1):016107, 2011.
[2] P. K. Gopalan, C. Wang, and D. Blei. Modeling overlapping communities with node popularities. NIPS 2013.
[3] A. Todeschini, X. Miscouridou and F. Caron. Exchangeable Random Measures for Sparse and Modular Graphs with Overlapping Communities. CoRR 2016.
[4] J. Lee, C. Heakulani, Z. Ghahramani, L. F. James, and S. Choi. Bayesian inference on random simple graphs with power law degree distributions. ICML 2017.
[5] P. Orbanz. Subsampling large graphs and invariance in networks. CoRR 2017.
[6] M. Zhou. Infinite edge partition models for overlapping community detection and link prediction. AISTATS 2015

---

### Official Review · AnonReviewer3 · 2018-11-02
**Application of GCN for overlapping community detection**

**Rating:** 3
**Confidence:** 5

**Review:**

This paper presents an overlapping community detection method. The idea is to use a graph neural network (namely, the graph convolutional network) with node embeddings constrained to be non-negative. The non-negative embeddings helps to learn the community membership of each node (and each node can belong to multiple communities).

The idea is natural, though not novel. The only main novelty, as compared to various other recently proposed graph embedding approaches, lies in making the node embeddings non-negative. Rest of the pieces are fairly standard, including the link functions, such as Bernoulli-Poisson.  Therefore the paper is quite thin in technical novelty.

In addition to the limited technical novelty, I have a few other concerns as well, including some on the experimental evaluation:

- Real-valued node embeddings obtained from shallow/deep graph embedding methods can be used with *overlapping* versions of k-means. This can be a solid baseline.

- The paper relies on subsampling the edges and non-edges to speed-up optimization. However, the encode still seems to use the entire adjacency matrix. If that is not the case, please clarify.

- The reported results are only on overlapping community detection. Most of the shallow/deep graph embedding methods can also be used for link prediction task (many of the recent paper report such results). It will be nice to provide results on this task.

- There has been some recent work on using deep generative models for overlapping community detection with node side information. For example, see "Deep Generative Models for Relational Data with Side Information" (Hu et al, 2017). Interestingly, they too use Bernoulli-Poisson link (but not GCN).

- None of the baselines are deep learning methods. As I pointed out, one can use real-valued embeddings from such methods with overlapping k-means (or other overlapping clustering methods). Link-prediction results can also be compared.

In summary, I think the paper lacks both in terms of technical novelty as well as experimental evaluation and therefore doesn't seem to be ready. I would encourage the authors to consider the suggestions above.

---

### Official Review · AnonReviewer2 · 2018-11-05
**Application of GNN to overlapping community detection**

**Rating:** 5
**Confidence:** 4

**Review:**

The current paper considers the overlapping community detection problem and suggests to use the so-called graph neural networks for its solution.

The approach starts from BigCLAM model and suggests to parametrize factor matrices (or embedding vectors) via neural network with graph adjacency matrix and node attributes as inputs. The obtained algorithm is tested on several datasets and its reported performance is superior to competitors.

This paper basically tries to introduce the dependence between embedding vectors for graph nodes, which recently became de facto standard approach in machine learning for graphs. The paper is very well aligned with recent literature on ML for graphs, which is focused on combining different ideas of deep learning, tailoring them to particular graph problem and reporting results on some datasets. Unfortunately, very rarely interesting new ideas appear in these papers, and current paper is not an exception.

I apologize for such a pessimistic view, but I don't see the results significantly interesting for the ICLR community and don't recommend acceptance. Some additional algorithmic/computational/theoretical insights are needed.

I have couple of minor issues to discuss:
1. For the sake of generality, I would recommend to use the general formula instead of particular 3-layer case in equation 3.
2. I don't think that it is really appropriate to call 3-layer model a 'deep learning model', I would recommend to just name it 'neural network'

Also, I think that experimentally paper is pretty strong, but it would be nice to see the repository with algorithm code and experiments available.

---

### Meta-Review · Area_Chair1 · 2018-12-01
**Interesting combination of existing techniques, with questions remaining be answered on modeling choices and experimental evaluations.**

**Confidence:** 5
**Recommendation:** Reject

**Metareview:**

The paper provides an interesting combination of existing techniques (such as GCN and and the Bernoulli-Poisson link) to address the problem of overlapping community detection. However, there were concerns about lack of novelty, evaluation metrics, and missing comparisons with previous work. The authors did not provide a response to address these concerns.